# SINR: Spline-enhanced implicit neural representation for multi-modal registration

**Vasiliki Sideri-Lampretsa**[1,2]                    VASILIKI.SIDERI-LAMPRETSA@TUM.DE
**Julian McGinnis**[1,2]                                   JULIAN.MCGINNIS@TUM.DE
**Huaqi Qiu**[1]                                               HUAQI.QIU@TUM.DE
**Magdalini Paschali**[4]                          MAGDA.PASCHALI@STANFORD.EDU
**Walter Simson**[4]                                     WSIMSON@STANFORD.EDU
**Daniel Rueckert**[1,2,3]                          DANIEL.RUECKERT@TUM.DE

[1] *Institute for AI in Medicine, Technical University of Munich, Germany*

[2] *Klinikum rechts der Isar, Munich, Germany*

[3] *Biomedical Image Analysis Group, Department of Computing, Imperial College London*

[4] *Department of Radiology, School of Medicine, Stanford University, USA*

**Editors:** Accepted for publication at MIDL 2024

## Abstract

Deformable image registration has undergone a transformative shift with the advent of deep learning. While convolutional neural networks (CNNs) allow for accelerated registration, they exhibit reduced accuracy compared to iterative pairwise optimization methods and require extensive training cohorts. Based on the advances in representing signals with neural networks, implicit neural representations (INRs) have emerged in the registration community to model dense displacement fields continuously. Using a pairwise registration setup, INRs mitigate the bias learned over a cohort of patients while leveraging advanced methodology and gradient-based optimization. However, the coordinate sampling scheme makes dense transformation parametrization with an INR prone to generating physiologically implausible configurations resulting in spatial folding. In this paper, we introduce SINR - a method to parameterize the continuous deformable transformation represented by an INR using Free Form Deformations (FFD). SINR allows for multi-modal deformable registration while mitigating folding issues found in current INR-based registration methods. SINR outperforms existing state-of-the-art methods on both 3D mono- and multi-modal brain registration on the CamCAN dataset, demonstrating its capabilities for pairwise mono- and multi-modal image registration.

**Keywords:** implicit neural representations, image registration, multi-modal

## 1. Introduction

Image registration involves aligning corresponding semantic regions in two or more images acquired with different imaging modalities or at separate points in time (Sotiras et al., 2013). In medical imaging, registration is vital for the quantitative interpretation of multiple images of the same patient, e.g., multi-modal image fusion, motion correction, and disease progression tracking. Conventional registration methods rely on pairwise instance optimization to learn gridded displacement fields, where a dissimilarity measure over a space of transformations is iteratively minimized (Ashburner, 2007; B. et al., 2005; Rueckert et al., 1999).

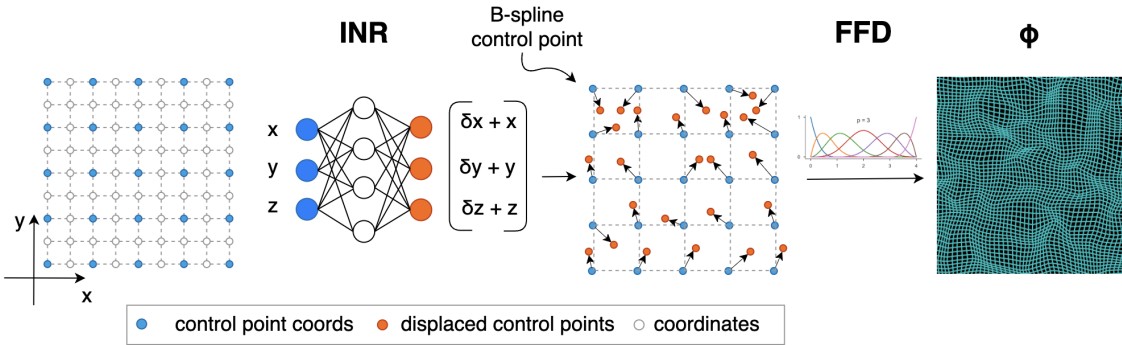

Figure 1: Given a densely sampled MRI, our approach, SINR, selects a subset of coordinates as control points to train an INR. Using gradient descent, the INR learns to model the continuous displacement field. By incorporating Free Form Deformations (FFD), we implicitly regularize the INR, achieving smoother transformations.

Data-driven registration-learning methods (Haskins et al., 2020), commonly implemented with CNNs, learn image correspondence over a dataset of image pairs and predict dense displacement fields (Balakrishnan et al., 2019), diffeomorphisms (Dalca et al., 2018; Mok and Chung, 2020) or parameters of the transformation model (Qiu et al., 2021). Due to training bias, these approaches are often resolution-dependent and can fail to generalize to other modalities. This registration-learning paradigm can offer fast inference time at the cost of accuracy (Hansen and Heinrich, 2021).

Recently, coordinate-based implicit neural representations (INRs) have been proposed to encode signals such as images or transformations as a function stored in the weights of a multi-layer perceptron (MLP) (Sitzmann et al., 2020; Tancik et al., 2020; Mildenhall et al., 2021). These approaches allow for a *continuous* representation of the underlying signal with potentially lower storage requirements than gridded representations (Dupont et al., 2021). Activation functions, such as ReLUs, sinusoids (SIRENs), and Gaussian activations, have been proposed (Rahaman et al., 2019; Sitzmann et al., 2020), with benefits to fidelity and training speed. INRs have also been used for registration; (Wolterink et al., 2022) proposed representing a dense deformable transformation between lung Computer Tomography (CT) images using INRs, while (Han et al., 2023) introduces mono-modal diffeomorphic registration with INRs. Finally, (Byra et al., 2023) examines the efficacy of INRs in improving the registration of mono-modal brain images in MRI. In these works, INRs are fitted only with the normalized coordinates of a single image pair and predict the dense displacement field, minimizing a conventional intensity-based dissimilarity measure. Additionally, unlike CNNs, INRs do not require a large training dataset.

INR-based registration methods commonly leverage sinusoidal activations, which can better represent higher frequency signal components (Sitzmann et al., 2020). However, the expressiveness of SIRENs is explicitly controlled by the sinusoids' frequency term $\omega$. Large $\omega$ values can lead to spatial folding and require higher explicit regularization to

enforce smoothness (Byra et al., 2023). These explicit regularization terms could negatively influence registration's accuracy.

The coordinate sampling might also affect the convergence and registration performance in the context of INR registration. To mitigate this negative effect due to random sampling, (Wolterink et al., 2022) suggested using a mask to ensure that regions of interest are more frequently sampled. This issue of sample prioritization becomes more pronounced in multi-modal registration where information-based metrics such as (normalized) mutual information (NMI) (Studholme et al., 1999; Wells et al., 1996) are employed. NMI is computationally expensive because it uses histograms to approximate the joint intensity distribution and, therefore, requires a large batch of coordinates for a representative distribution of the image content for successful registration. Consequently, the computational resources scale quickly with image size and dimensionality, reducing tractability with scale. Conversely, smaller batch sizes have been shown to lead to higher signal modeling accuracy in INRs and increased training stability (McGinnis et al., 2023), thus posing a challenge to the NMI-based multi-modal registration.

To address the limitations of current implicit registration methods, we propose Spline-enhanced INR (SINR), which parameterizes the implicit representation of a deformable transformation using Free Form Deformations (FFD) (Rueckert et al., 1999). FFD, originally proposed for the flexible manipulation of 3D shapes, deforms a control lattice, allowing the implicit regularization of SINR to produce smoother transformations without compromising registration accuracy. Further, the FFD model reduces the sensitivity of SINR to the choice of frequency ($\omega$) in the SIREN activation. SINR only parameterizes spatially sparse FFD control points, reducing the computational burden of coordinate sampling. Unlike previous work (Wolterink et al., 2022), this allows SINR to use NMI efficiently with INRs not only for mono-modal but also for multi-modal registration.

Our contributions are the following:

- We propose SINR, a registration method that parameterizes deformable transformation by combining implicit neural representation (INR) with free-form deformation (FFD). The efficient spatial sampling and intrinsic smoothness, benefits of the FFD model, lead to improved optimization and state-of-the-art registration performance;
- SINR exploits the FFD control point sparsity to efficiently calculate NMI, which enables multi-modal INR-based registration for the first time;
- SINR achieves accurate registration with comparable or improved transformation regularity. We evaluate registration performance on mono-modal and multi-modal brain MRIs and compare it with iterative and learning-based methods.

## 2. Method

### 2.1. Pairwise Image Registration

Given two $n$-dimensional images, a fixed image $F$ and a moving image $M$ with $F, M : \Omega \subset \mathbb{R}^n \to \mathbb{R}$ ($n = 3$ for 3D MRIs), image registration aims to find an optimal spatial transformation $\phi : \mathbb{R}^n \to \mathbb{R}^n$ such that the transformed moving image is most similar to the fixed image. Typically, this is formulated as an optimization problem $\phi^* = \arg\max_\phi \mathcal{J}(F, M, \phi)$

where the distance between the images is minimized with constraints on the transformation. We denote the objective function $\mathcal{J}$ as:

$$\mathcal{J}(F, M, \phi) = \mathcal{D}(F, M \circ \phi) + \lambda \mathcal{R}(\phi), \tag{1}$$

where $\mathcal{D}$ is an intensity dissimilarity measure and $\mathcal{R}$ is the regularization on the transformation field whose effect is controlled by the parameter $\lambda$.

## 2.2. Free Form Deformations

Free-form deformations involve the flexible alteration of images by adjusting control points within a parametric space, allowing non-rigid transformations. B-spline-based FFD models parametrize a deformable transformation between two images by defining a mesh of control points in the spatial domain of the image volume (Rueckert et al., 1999). Giving a uniform spacing $\delta$, the FFD can be formulated as:

$$\mathsf{u}(x, y, z) = \sum_{l=0}^{3} \sum_{m=0}^{3} \sum_{n=0}^{3} B_l(u) B_m(v) B_n(w) c_{i+l} c_{j+m} c_{k+n}, \tag{2}$$

where $(i, j, k)$ are the indices of the control point which is closest to the origin in the control point cube that encloses $(x, y, z)$, $B$ are the B-spline basis functions as presented in (3), and $(u, v, w)$ are the normalized local coordinates of $(x, y, z)$ in its enclosing control point cube.

$$B_0(u) = \frac{(1-u)^3}{6}, B_1(u) = \frac{3u^3 - 6u^2 + 4}{6}, B_0(u) = \frac{u^3}{6} \tag{3}$$

This transformation parametrization has some advantages due to its B-spline formulation. Firstly, B-splines have local support, which makes them a compelling choice for parameterizing deformable transformation. In other words, each control point affects the transformation only in its local neighborhood. Moreover, the resolution of the control point mesh is proportional to the transformation smoothness. Larger spacing between control points facilitates the representation of more global, smoother, nonrigid deformations, while smaller spacing allows for the modeling of highly localized nonrigid deformations. Therefore, the control point spacing can implicitly act as a method constraint, promoting smoothness.

## 2.3. Proposed Method - SINR

While conventional iterative methods estimate the transformation using pairwise optimization and CNN-based methods learn the transformation over a cohort, we employ a coordinate-based INR with the transformation between a pair of images due to its high signal fidelity and fast training speed. Leveraging the compressed representation, we propose to exploit the inherent smoothness of the B-spline FFD we previously described and use an INR $f_\theta$ with trainable parameters $\theta$ to approximate the transformation $\phi(\boldsymbol{x_{cp}}) = \boldsymbol{x_{cp}} + u(\boldsymbol{x_{cp}})$ between a given pair of images $F, M$, where $\boldsymbol{x_{cp}} \in \Omega$ are the control point coordinates and $\phi(\boldsymbol{x}) = f_\theta(\boldsymbol{x})$ are the displacements on the control point coordinates. The $L$-layer network is modelled as $\mathbf{f_\theta} = \mathbf{f_L} \circ \mathbf{f_{L-1}} \circ \ldots \circ \mathbf{f_1}$, with

$$\mathbf{h}_l = f_l(\mathbf{h}_{l-1}) = \psi(W_l \mathbf{h}_{l-1} + b_l), 0 \le l \le L, \tag{4}$$

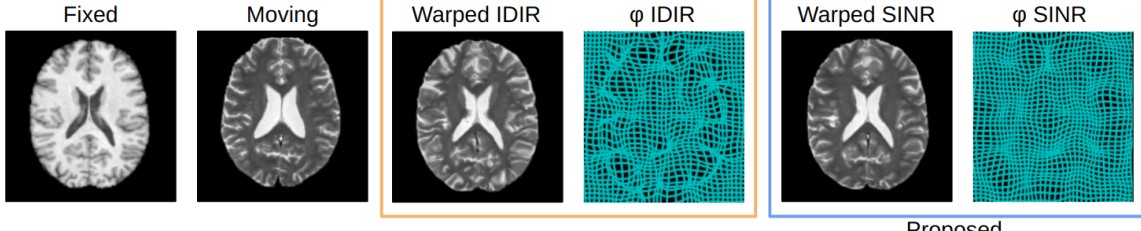

Figure 2: Qualitative results on T1w-T2w registration. The proposed SINR with SIREN activations achieves more plausible results (0.51% folding ratio) compared to IDIR with SIREN (Wolterink et al., 2022) activation (0.87% folding ratio).

where $\mathbf{W_1}$ denote the weights, $\mathbf{b_l}$ the bias, $\mathbf{h}_l$ the hidden feature vector for the $l$-th layer, and $\psi$ the network's activation function, where we experiment with both, ReLU and SIREN.

## 3. Experimental Setup

**Evaluation metrics.** Assessing the registration performance can be challenging since the ground truth deformations are unknown. Therefore, registration accuracy and regularity are evaluated with surrogate measures. Accuracy is determined by assessing the overlap between the anatomical segmentation using the Dice score. The regularity of transformation is evaluated based on the Jacobian determinant. The extent of *folding* in the image due to the transformation is measured by the percentage of points with $\mathcal{J} = |\nabla\phi| < 0$.

**Datasets.** We evaluate our work on the inter-subject brain registration using the Cam-CAN[1] dataset (Shafto et al., 2014), (Taylor et al., 2017). The dataset consists of 310 T1w and T2w MR 3D volumetric images of size $192 \times 192 \times 192$ and 1mm$^3$ isotropic spatial resolution, which we split into 80% training, 10% validation and 10% test set. We normalize all images to the MNI space (Horn, 2016) using affine registration, ensuring an isotropic spatial resolution with a voxel size of 1mm$^3$. We perform skull-stripping using ROBEX (Iglesias et al., 2011) and bias-field correction with SimpleITK (Lowekamp et al., 2013). For assessment purposes, we obtained automated segmentation of 138 cortical and subcortical structures, categorized into 5 groups, using MALPEM (Ledig et al., 2015).

**Baselines.** We first compare the proposed method to a conventional iterative method Medical Image Registration ToolKit (MIRTK) (Schuh et al., 2014), which is based on the FFD model. We also compare against two CNN-based deep learning methods; the widely used Voxelmorph (VMorph) (Dalca et al., 2018), which outputs a dense displacement field, and Modality-Invariant Diffeomorphic Deep Learning Image Registration (MIDIR) (Qiu et al., 2021), which predicts FFD as transformation. Furthermore, we compare against an INR-based method named Implicit Neural Representations for Deformable Image Registration (IDIR) (Wolterink et al., 2022), which outputs a dense displacement field instead of a

---

1. https://cam-can.mrc-cbu.cam.ac.uk/dataset/

Table 1: Best scores of SINR and its competitors. The mean and std of the Dice score over anatomical structures are reported along with the transformation's folding ratio and whether or not the method utilizes the FFD transformation.

| Method | FFD | T1w-T1w CamCAN | | T1w-T2w CamCAN | |
|---|---|---|---|---|---|
| | | Dice $\pm$ std $\uparrow$ | Folding % $\downarrow$ | Dice $\pm$ std $\uparrow$ | Folding % $\downarrow$ |
| Affine | n/a | $0.619 \pm 0.01$ | - | $0.619 \pm 0.01$ | - |
| MIRTK | ✓ | $0.833 \pm 0.02$ | 0.11 | $0.755 \pm 0.01$ | 0.14 |
| VMorph [CNN] | ✗ | $0.812 \pm 0.06$ | 0.31 | $0.733 \pm 0.04$ | 0.19 |
| MIDIR [CNN] | ✓ | $0.817 \pm 0.06$ | 0.23 | $0.735 \pm 0.04$ | 0.12 |
| IDIR [ReLU-MLP] | ✗ | $0.806 \pm 0.02$ | 0.44 | $0.683 \pm 0.03$ | 0.15 |
| **SINR** [ReLU-MLP] | ✓ | $0.789 \pm 0.03$ | 0.38 | $0.721 \pm 0.06$ | 0.05 |
| IDIR [SIREN] | ✗ | $0.837 \pm 0.05$ | 0.84 | $0.736 \pm 0.02$ | 0.81 |
| **SINR** [SIREN] | ✓ | $\mathbf{0.855 \pm 0.06}$ | 0.59 | $\mathbf{0.784 \pm 0.04}$ | 0.27 |

parameterized transformation. We also test this approach with both SIRENs and ReLUs as activation functions.

**Implementation.** We trained all the mono-modal experiments with Normalized Cross Correlation and the multi-modal ones using differentiable NMI (De Vos et al., 2019) as image similarity measure. All baselines and the proposed method incorporate bending energy for regularization as introduced by (Rueckert et al., 1999). All the INRs were trained using the ADAM optimizer with a $10^{-4}$ learning rate for a maximum of 2500 epochs. We considered sampling the coordinates inside the brain mask only for the baseline INRs and not for SINR. The dense mono-modal experiments used a coordinate batch size of 10k samples, while the multi-modal experiments used a batch size of 890k ($\frac{1}{8}$ of the total points) to ensure convergence with NMI. We tune the selection of hyperparameters, namely the regularization weight $\lambda$ and $\omega$, by evaluating every 50 steps and performing an early stopping if the folding ratio becomes larger than 0.9%. This threshold is chosen empirically by evaluating the registration performance qualitatively. We refer the reader to Appendix A, Figure 5, where the resulting transformation demonstrates approximately 0.9% folding. We choose the hyperparameters that achieve the highest Dice score for every method while not surpassing this folding ratio threshold. For the dataset, optimal outcomes are observed when the control points are spaced at 2mm$^3$. The code is publicly available[2].

## 4. Results and Discussion

In our experiments, SINR, using sinusoidal activation functions, achieves the highest registration accuracy in Dice, surpassing the INR-based IDIR by 1.8% in the mono-modal and 4.8% in the multi-modal case. SINR performs sparse spatial sampling with its use of FFD control points. This sparse sampling strategy allows for stable computation of NMI via more efficient spatial sampling over the entire domain compared with random sampling of

---

2. https://github.com/vasl12/SINR.git

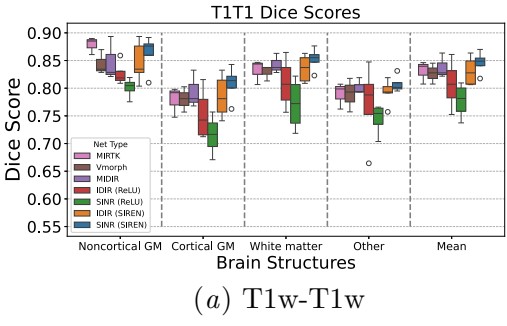
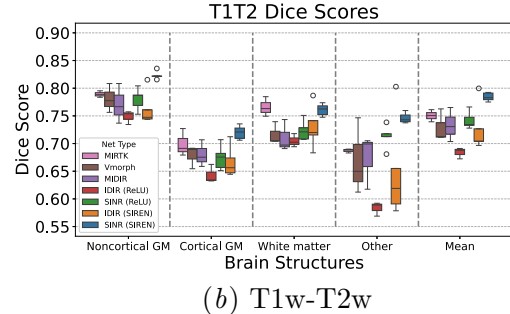

(a) T1w-T1w    (b) T1w-T2w

Figure 3: Dice scores for brain registration by structure, indicating the average (mean) across White Matter (WM) and Grey Matter (GM) areas. A larger version can be found in Appendix A, Figure 6.

IDIR. Compared to IDIR, SINR can achieve higher Dice without requiring masked sampling strategies. Similarly, it demonstrates superior performance over the conventional iterative method (MIRTK) by 2.2% and 2.9%, respectively. Moreover, the CNN-based VMorph and MIDIR baselines underperform compared to the pairwise SINR (by approximately 2% and 5% for mono- and multi-modal) because they are not optimized individually, but they estimate the transformation based on the prior learned over the whole training set. Additionally, our results highlight the superiority of SIREN-based methods over ReLU methods. SIRENs can represent signals with higher frequency components and, hence, more accurate transformations, while ReLUs tend to produce smoother transformations, which might not be descriptive enough and, as a result, lack performance. Comparing SINR with SIREN vs. with ReLU, SINR with SIREN achieves approximately 1% higher Dice, which shows that combining ReLUs with the FFD leads to over-smoothed transformations that lack the desired expressiveness.

Figure 3 demonstrates the accuracy of the SINR compared to the baselines for individual classes and the overall mean, confirming the finding of Table 1. SINR with SIREN demonstrates a superior mean Dice score and outperforms all baselines in almost all the individual classes in both registration tasks. MIRTK achieves a marginally higher Dice score for Noncortical GM in the mono-modal case and a comparable score in the multi-modal case for White Matter. However, the proposed method achieves substantially higher Dice for other structures in both mono- and multi-modal cases (c.f. Table 3). We further refer the reader to the Appendix A, Figure 6 for an enlarged version of Figure 3.

Regarding folding ratio, the proposed SINR with SIREN and ReLU activations manages to mitigate the folding ratio effect, which IDIR is prone to, as shown in Table 1. SINR equipped with ReLUs demonstrates the lowest folding ratio among all its competitors, making it a suitable candidate for applications in which a smooth transformation is desired, such as inhale-exhale lung registration. For multi-modal registration, our SIREN-based SINRs' folding ratio was marginally higher compared to other baselines but demonstrated an improved Dice score over them. Qualitatively this can also be confirmed by Figure 2, where it can be seen that the FFD results in a smoother, more accurate transformation suitable for brain registration in comparison to IDIR.

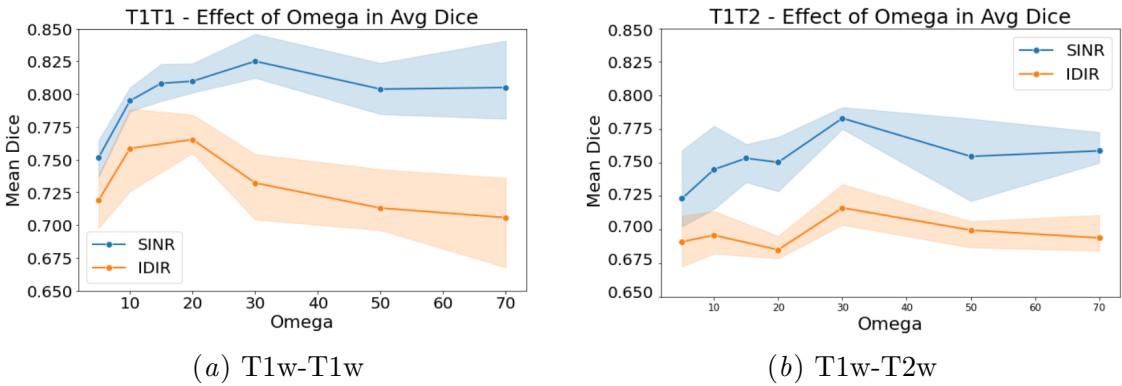

$(a)$ T1w-T1w                 $(b)$ T1w-T2w

Figure 4: Effect of $\omega$ on the Dice score for a fixed range of folding percentage of $\sim0.9\%$. SINR with SIREN activations outperforms SIREN-IDIR for all values of $\omega$ for mono-modal and multi-modal registration settings.

**Hyperparameter robustness:** We examine the influence of $\omega$ on registration accuracy while maintaining the folding ratio below 0.9% using the dense displacement INR method IDIR and the proposed SINR, which uses FFD. The results are presented in Figure 4 as the average and standard deviation of Dice scores over $\omega$ ranging from 5 to 70. SINR displays consistently higher Dice scores over all $\omega$ values, showing robustness to $\omega$ hyperparameter selection. Notably, for the T1w-T2w registration setting, both methods achieve the peak Dice score with an $\omega$ value 30, as proposed in (Sitzmann et al., 2020). In the mono-modal setting, SINR achieves the best results with an $\omega$ of 30, while IDIR requires an $\omega$ value of 20 for maximal performance. Low values of $\omega$ reduce Dice scores for both SINR and IDIR. The performance degradation over $\omega$ is more pronounced on IDIR and remains constant with larger $\omega$, while SINR displays reduced sensitivity to $\omega$ and stabilized performance for larger $\omega$ values.

## 5. Conclusion

In this work, we propose parameterizing the deformable transformation using an INR with Free Form Deformations. Through this combination, we benefit from the lightweight, fast-fitting INR and the inherent smoothness of B-spline FFD parametrization to achieve state-of-the-art performance in mono-modal and multi-modal brain registration. Extensive experimentation demonstrates the versatility of our approach, which not only outperforms conventional approaches, CNN methods, and dense INRs but also mitigates the implausible transformation percentage of the latter. We further perform an ablation study showing that the proposed FFD-enhanced INR is more robust against the activation function's frequency choice. Future work will extend this approach to other data modalities and anatomies, such as abdominal CT scans and further loss functions. Finally, an interesting future direction is to design an architecture that combines the registration with the FFD within the INR.

## 6. Acknowledgements

JM is supported by the Bavarian State Ministry for Science and Art (Collaborative Bilateral Research Program Bavaria – Québec: AI in medicine, grant F.4-V0134.K5.1/86/34).

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
