# OpenReview forum: "SINR: Spline-enhanced implicit neural representation for multi-modal registration"
_MIDL.io/2024/Conference — MIDL 2024 Oral_

### Official Review · Reviewer_b4Ub · 2024-02-26

**Confidence:** 5
**Preliminary Rating:** 3
**Final Rating:** 4

**Summary:**

This work parameterizes the deformable transformation using an INR with Free Form Deformations, benefiting from the lightweight, fast fitting from INR and the inherent smoothness from B-spline FFD parametrization.

The proposed method achieves state-of-art performance in mono-modal and multi-modal brain registration.

**Strengths:**

- Parameterizing deformable transformation by combining implicit neural representation with free-form deformation is interesting.
- The paper is well-organized and easy to follow.
- The ablation study shows that  FFD-enhanced SINR is more robust against the activation function frequency choice.

**Weaknesses:**

- Simply combining existing Free Form Deformations formulation with implicit neural representation is novelty-limited. As mentioned in related work, both are well explored in registration.
- MIND-loss and diffeomorphisms can solve the limitations of current implicit registration methods, such as inefficiency and smoothness. The authors should give more clear motivation for SIRN registration.
- In Table 1, the running time should be demonstrated. Additionally, IDIR [ReLU-MLP] shows higher accuracy than the proposed SINR [ReLU-MLP], indicating the inefficiency of the FFD-enhanced registration. The activation function choices seem to have a more important influence than the FFD-enhanced strategy. Please explain the reasons.

**Detailed Comments:**

- It would be better to add results about diffeomorphic registration methods, such as Diffeomorphic VMorph.
- The training and test time of learning-based registration and test time of conventional registration should be illustrated.
- The GitHub rep should add detailed ReadME.md to explain the usage.

**Justification Of Final Rating:**

The rebuttal has addressed concerns about novelty and unclear motivation and the manuscript has been improved.
I have raised my ranking to Weak Accept.  It is a solid paper, but nothing ground-breaking.

**Justification Of The Preliminary Rating:**

The main concerns about limited novelty, unclear motivation, and experiment verification.
- Simply combining existing Free Form Deformations formulation with implicit neural representation is novelty-limited. As mentioned in related work, both are well explored in registration.
- MIND-loss and diffeomorphisms can solve the limitations of current implicit registration methods, such as inefficiency and smoothness. The authors should give more clear motivation for SIRN registration.
- In Table 1, the running time should be demonstrated. Additionally, IDIR [ReLU-MLP] shows higher accuracy than the proposed SINR [ReLU-MLP], indicating the inefficiency of the FFD-enhanced registration. The activation function choices seem to have a more important influence than the FFD-enhanced strategy. Please explain the reasons.

**Questions To Address In The Rebuttal:**

Please address the concerns about novelty, motivation, and experiment verification.

- Simply combining existing Free Form Deformations formulation with implicit neural representation is novelty-limited. As mentioned in related work, both are well explored in registration.
- MIND-loss and diffeomorphisms can solve the limitations of current implicit registration methods, such as inefficiency and smoothness. The authors should give more clear motivation for SIRN registration.
- In Table 1, the running time should be demonstrated. Additionally, IDIR [ReLU-MLP] shows higher accuracy than the proposed SINR [ReLU-MLP], indicating the inefficiency of the FFD-enhanced registration. The activation function choices seem to have a more important influence than the FFD-enhanced strategy. Please explain the reasons.

---

> ### Author Response · Authors · 2024-03-17
>
> We would like to thank Reviewer b4Ub for providing insightful feedback and raising unclear questions. We address all concerns in the subsequent rebuttal and have revised the manuscript accordingly.
>
> > Simply combining existing Free Form Deformations formulation with implicit neural representation is novelty-limited. As mentioned in related work, both are well explored in registration.
>
> INRs, specifically INRs for registration, is an emerging field in deep learning. For this reason, exploring the combination of well-established registration methods (FFD) with novel deep-learning methods could benefit the field. The INR field as such is not yet thoroughly explored. When registering brain images we were presented by the umnet challenge of spacial folding in the context of INRs. In this work, we wanted to motivate the research on the regularisation of INRs for registration in terms of folding issues and highlight potential future research in this direction. We believe our work may be one of the first to explore this research area from one perspective, while there are likely other meaningful and interesting directions as well.
>
> > MIND-loss and diffeomorphisms can solve the limitations of current implicit registration methods, such as inefficiency and smoothness. The authors should give more clear motivation for SIRN registration.
>
> We appreciate the reviewer's suggestion to investigate the application of MIND-loss in the realm of image registration. We recognize the potential value this extension could bring to our work. However, it's essential to note that each multi-modal similarity or distance measure carries its own set of strengths and limitations. We acknowledge that there isn't a singular perfect measure capable of facilitating seamless multi-modal registration without encountering challenges. This far, Mutual Information (in various forms such as normalized, local, etc.) has been the most commonly utilized and robust measure for multi-modal registration. Consequently, we opted to conduct experiments using MI, but we will happily consider MIND in our future work.
> Selecting the optimal transformation to accurately represent the underlying deformation poses a significant challenge, primarily because image registration constitutes an inverse problem with the sought transformation being unknown a priori. Particularly in scenarios like inter-subject brain registration, the diffeomorphic transformation model may not be immediately suitable, as the sought transformation could display non-smooth behavior owing to the intricate nature of brain structures. In this study, we deliberately refrained from making this assumption. Nonetheless, we acknowledge that diffeomorphisms could enhance overall performance in various applications or modalities.
>
> > In Table 1, the running time should be demonstrated. Additionally, IDIR [ReLU-MLP] shows higher accuracy than the proposed SINR [ReLU-MLP], indicating the inefficiency of the FFD-enhanced registration. The activation function choices seem to have a more important influence than the FFD-enhanced strategy. Please explain the reasons.
>
> The selection of activation functions remains an active area of investigation within the realm of INRs [i, ii]. Specifically, ReLUs are known for their inability to capture high-frequency details, resulting in smoother and potentially "blurrier" representations. In this study, we included comparisons with INRs employing ReLUs, both for baseline and SINR cases, to provide a comprehensive analysis. However, we contend that INRs utilizing ReLUs may not be well-suited for inter-subject brain registration, and incorporating FFD cannot fully mitigate the drawbacks associated with INRs. Conversely, INRs employing sinusoidal activation functions emerge as a suitable choice for image registration. In this scenario, it becomes evident that integrating FFD not only enhances INR registration accuracy but also improves registration regularity (folding ratio).
> (i) Sitzmann, Vincent, et al. "SIREN" NeurIPS2020
> (ii) Tancik, Matthew, et al. "Fourier Features." NeurIPS2020
>
> > The training and test time of learning-based registration and test time of conventional registration should be illustrated.
>
> Thank you for this suggestion, please refer to Table 2 in the updated manuscript.
>
> > It would be better to add results about diffeomorphic registration methods, such as Diffeomorphic VMorph.
>
> As we do not assume diffeomorphism, we feel that the comparison with these methods would not be fair, as we would also need to enforce these conditions on INRs, which is a new direction of research as such. This is very interesting, and we aim to investigate these ideas in the future!
>
> > The GitHub rep should add detailed ReadME.md to explain the usage.
>
> Thank you for the suggestion, we have adapted the Readme.md accordingly.

---

> > ### Comment · Reviewer_b4Ub · 2024-03-19
> > **Change my score from 3: Borderline to 4: Weak accept**
> >
> > Concerns about novelty and unclear motivation have been addressed.

---

### Official Review · Reviewer_n1vA · 2024-02-28

**Confidence:** 5
**Preliminary Rating:** 4
**Final Rating:** 5

**Summary:**

The authors introduce a novel multi-modal non-linear volumetric registration framework. It is a spline-enhanced INR (SINR) solution that implements the classic FFD registration framework. This allows for both a uni-modal and multi-modal problem setup (via the use of NMI implementation for the latter).

The submission is well-written and well-motivated. Use of INR for non-rigid registration in both mono-modal and multimodal registration. The FFD model helps with reducing folding issues compared to INR (but not with other CNN and traditional methods).

The authors committed to sharing their code on GitHub.

**Strengths:**

The authors implement a classic non-linear registration solution in a DL framework. This allows for efficient and highly accurate outcomes compared to some SOTA methods. The here presented solution allows for solving both uni-modal and multi-modal registration problems. Even though the experiments are run on a relatively small dataset, the results seem promising.

**Weaknesses:**

It would have been great if more details are provided on the segmentation labels that were used for the performance evaluation.
It would have been great if the authors were more clear / explained it in more detail, whether their higher folding percentage compared to the other solutions is of a concern.

**Detailed Comments:**

How generalizable is the performance of this method on data sets other than the one tested? VMorph was, as far as I know, tested on many different data sets.

Some details are missing regarding Dice computation (Was an average or a global Dice measure computed? -- Mean over all structures? Sum of Dices?). What is the Dice computed over -- what are the "anatomical structures"? Were there many labels segmented or just the same larger ones as they appear on the performance plots? Is this an average Dice score? Regarding the quantitative experimental evaluation, under what ROIs are the metrics computed? For example, what falls under "Other"?

How generalizable is the tool to non-CamCAN data sets? Eval was done on N = 31, if I understand.

It would have been great if the limitation of high folding compared to the other methods was discussed. Especially in the T1T1 experiments, where this number is half in the non-IDIR methods. The authors, however, describe it as "comparable".

**Justification Of Final Rating:**

The revised additions have answered my (and the other reviewers') questions and the submission improved. Details about the experimental setup and the folding ratio optimization have been provided. I increased my score to strong accept even with the limited experimental data presented.

**Justification Of The Preliminary Rating:**

There is a lack of detail in describing the experimental evaluation  (re Dice score, segmentation labels).
The experimental outcome is promising, but it is achieved on a limited data set. The performance (esp folding %) is not fully discussed.

**Questions To Address In The Rebuttal:**

Can you provide further details on the segmentation labels that were used for the performance evaluation?
Can you explain in more detail whether the higher folding percentage of SINR compared to the other solutions is of a concern?

---

> ### Author Response · Authors · 2024-03-17
>
> We would like to thank Reviewer n1vA for providing insightful feedback and interesting questions. We address all concerns in the subsequent rebuttal and have revised the manuscript accordingly.
>
> > How generalizable is the performance of this method on data sets other than the one tested? VMorph was, as far as I know, tested on many different data sets.
> > How generalizable is the tool to non-CamCAN data sets? Eval was done on N = 31, if I understand.
>
> The proposed approach was solely assessed using T1w-T1w brain registration and T1w-T2w brain registration within the CamCAN dataset. Nonetheless, given that our methodology isn't specifically tailored to these modalities within this dataset, we are confident in its potential to generalize across diverse datasets, which we will assess, together with other ideas from the reviewing process, in future work.
>
> > Some details are missing regarding Dice computation (Was an average or a global Dice measure computed? -- Mean over all structures? Sum of Dices?). What is the Dice computed over -- what are the "anatomical structures"? Were there many labels segmented or just the same larger ones as they appear on the performance plots? Is this an average Dice score? Regarding the quantitative experimental evaluation, under what ROIs are the metrics computed? For example, what falls under "Other"?
>
> > Can you provide further details on the segmentation labels that were used for the performance evaluation? Can you explain in more detail whether the higher folding percentage of SINR compared to the other solutions is of a concern?
>
> We would like to thank the reviewer for highlighting the absence of details regarding the segmentation labels. To assess our work, we acquired automated segmentation of 138 cortical and subcortical structures, grouped into 5 categories, utilizing MALPEM [i]. More information about the labels can be found at http://christianledig.com/Material/MALPEM/m100_Report.pdf. We have updated the manuscript to incorporate this previously omitted information. Concerning the DICE score, we employ the standard computation method. If reviewer n1vA deems it necessary, we will include its formula in the Appendix.
>
> (i) Ledig, Christian, et al. "Robust whole-brain segmentation: application to traumatic brain injury." MedIA (2015).
>
> > It would have been great if the limitation of high folding compared to the other methods was discussed. Especially in the T1T1 experiments, where this number is half in the non-IDIR methods. The authors, however, describe it as "comparable".
>
> Balancing the folding ratio and registration accuracy entails a trade-off, especially in the challenging scenarios of inter-subject registration where the folding cannot be avoided. Decreasing the folding ratio can be achieved through heightened regularization, which occasionally compromises accuracy. Employing weak deformation regularization permits meticulous alignment by accommodating anatomically implausible deformation magnitudes and irregularities. Conversely, intense regularization yields diminutive and more centralized deformations but yields suboptimal alignment. We choose early stopping criteria and an optimal regularization weight to maintain an acceptable folding ratio, which is why we referred to it as comparable. However, we recognize that the folding ratio tends to be higher in INR methods. Consequently, we have revised the manuscript to reflect this observation.

---

> > ### Comment · Reviewer_n1vA · 2024-03-26
> >
> > Thank you for the authors' clarifications. The revised additions have answered my questions and the submission improved.

---

### Official Review · Reviewer_8syk · 2024-02-29

**Confidence:** 5
**Preliminary Rating:** 5
**Recommendation:** Oral
**Final Rating:** 5

**Summary:**

This work proposes a registration method that merges traditional free form deformations with INR registration. Rather than parameterizing a deformation or velocity field directly, the INR is used to parameterize the grid of control points for B-splines. The method is evaluated on the Cam-CAN dataset compares favourably with sensible baselines.

**Strengths:**

The paper is well-written and easy to follow. While a few details on the method are missing, the availability of the code makes this much less of a problem.

The results are presented in a clear and structured way, and show favourable comparisons with sensible baselines.

**Weaknesses:**

The experiments are performed with the same number of iterations, but the batch size of the presented method is much larger in the mono-modal experiments. Hence, that network saw far more samples than the baseline, meaning for some images it may have converged while the baseline had not yet. Conversely, in the multi-modal experiments, the batch size of the baseline is much higher than what is typically used (i.e. 3.5M instead of 10k), which may have had a detrimental effect on the optimisation process.

The main drawback of the baseline INR method is its slow inference compared to other modern registration methods; the paper makes no mentions of runtime, which I feel would be valuable to better understand the potential applications.

It is somewhat unclear what data the method was tested on. The authors specify there was a train/validation/test split of 80/10/10, but unless I've misunderstood, the method does not require training. Was 80% of the dataset used for hyper-parameter tuning? This seems excessive.

**Detailed Comments:**

I may have missed it, but I do not think the paper mentions how batches are sampled in the presented method. Looking at the code, the answer seems to be "use everything all the time"; this is a significant difference from the baseline INR method, so it should be mentioned.

The paper does not mention whether masks were used for the baseline experiments. Were coordinates sampled from a foreground mask, or were they sampled uniformly in the images?

**Justification Of Final Rating:**

My assessment remains the same: the paper presents a strong, well explained registration method and the experiments show the method is beneficial compared to sensible baselines. My main concerns have been addressed appropriately, and I look forward to discussing the work with the authors in more detail.

**Justification Of The Preliminary Rating:**

The paper presents a strong, well explained and intuitive registration method. The experiments show the method is beneficial compared to sensible baselines. Additionally, the approach is not constrained to any specific domain, making it possibly useful for a large number of different registration problems.

**Questions To Address In The Rebuttal:**

If I understand correctly, in the presented method, the authors fully sample their space of control points each iteration. While this is evidently effective, it makes me question what the INR adds to the original FFD formulation in this setting. INRs are useful for latent interpolation, arbitrarily precise derivatives for regularization and compact representation of deformation fields, but none of these properties is being exploited here. I would expect the results to be very similar to the MIRTK results. While the results are reasonably close, the method does clearly outperform MIRTK. The paper should discuss what the advantage of the proposed method is compared to classical FFD; is this merely better hyper-parameter tuning? Or is there another intuitive reason why SINR can consistently work better?

The main drawback of registration with INRs is the inference time, which is generally slower than both learning-based methods and GPU implementations of traditional methods. How does the presented method compare to the INRs by Wolterink et al. in terms of runtime? How do the performance over time graphs look compared to the baseline INRs? It would be nice to see this, possibly in an appendix.

I'm worried the 3.5M batch size for the multi-modal experiments may have had a detrimental effect on the optimisation process, which may have a larger effect than mild instability of the mutual information computation. I would expect 100k would be more than enough to get a representative distribution. Did you try this with multiple batch sizes to confirm 3.5M is the best option?

Please clarify what is meant by the train/validation/test split of 80/10/10. There is no training in the "classical" sense, so which parts of the dataset were used for what, exactly?

---

> ### Author Response · Authors · 2024-03-17
>
> We would like to thank Reviewer 8syk for providing insightful feedback and interesting questions. We address all concerns in the subsequent rebuttal and have revised the manuscript accordingly.
>
> >The experiments are performed with the same number of iterations, but the batch size of the presented method is much larger in the mono-modal experiments. [...]
>
> >I'm worried the 3.5M batch size for the multi-modal experiments may have had a detrimental effect on the optimisation process, which may have a larger effect than mild instability of the mutual information computation. I would expect 100k would be more than enough to get a representative distribution. Did you try this with multiple batch sizes to confirm 3.5M is the best option?
>
> We appreciate the reviewer's comment. Firstly, we want to clarify that we do not utilize the same number of iterations for all experiments and we revise the manuscript accordingly. Instead, for each image pair the number of iterations is determined by the stopping criteria. This approach ensures that the baseline has been exposed to no fewer points than the SINR.
> Employing a larger number of points in the multi-modal case may result in a slight decrease in optimization speed; however, we have not detected any detrimental effects on the optimization process. This number of points ensures accurate and robust results. Nevertheless, we acknowledge that conducting further experiments on the batch size would be advantageous to investigate its impact on the optimization process.
>
> >The main drawback of the baseline INR method is its slow inference compared to other modern registration methods; the paper makes no mentions of runtime, which I feel would be valuable to better understand the potential applications.
>
> >[...] How does the presented method compare to the INRs by Wolterink et al. in terms of runtime? How do the performance over time graphs look compared to the baseline INRs? It would be nice to see this, possibly in an appendix.
>
> Thank you for this suggestion, please refer to Table 2 in the updated manuscript.
>
> >Please clarify what is meant by the train/validation/test split of 80/10/10. There is no training in the "classical" sense, so which parts of the dataset were used for what, exactly?
>
> Deep learning CNN networks for registration, are trained on a cohort basis, whereas conventional methods like MIRTK and INR-based approaches are fitted on a pair of subjects. We opted to split the dataset into 80% for training, 10% for validation, and 10% for testing. Subsequently, we trained the CNNs on the training set and fine-tuned their hyperparameters using the validation set. The test set evaluated the networks' ability for accurate and robust registration. Unlike CNN networks, pairwise methods do not require a training/validation/test split, and their hyperparameters can be tuned pairwise. Nevertheless, for comparative purposes against CNN methods, we calibrated the hyperparameters on the validation set and evaluated registration performance using the same test set as mentioned earlier.
>
> >I may have missed it, but I do not think the paper mentions how batches are sampled in the presented method. Looking at the code, the answer seems to be "use everything all the time"; this is a significant difference from the baseline INR method, so it should be mentioned.
>
> >[...] Were coordinates sampled from a foreground mask, or were they sampled uniformly in the images?
>
> We would like to thank the reviewer for bringing this omission to our attention and we revise the manuscript accordingly. For the baseline INR, points are randomly sampled within a brain mask, which we empirically observe enhances the performance and convergence speed. In the case of SINR, we instead sample only a subset of points at intervals of 2mm^3 spacing on a grid (⅛ of the total points). During inference, all points are considered.
>
> >[...] the authors fully sample their space of control points each iteration. While this is evidently effective, it makes me question what the INR adds to the original FFD formulation in this setting. [...] While the results are reasonably close, the method does clearly outperform MIRTK. The paper should discuss what the advantage of the proposed method is compared to classical FFD; [...]
>
> The proposed SINR method samples 1/8 of all points as control points at each iteration. This approach allows it to retain the advantages of INR, as the optimization occurs exclusively within the INR framework. The FFD is incorporated post-INR fitting, leveraging its demonstrated capability to alleviate significant spatial folding issues, to which baseline INRs are particularly susceptible. In essence, the SINR method combines the best of both worlds: utilizing a neural network for the fitting process and FFD to ensure smooth and realistic transformations. Additionally, SINR outperforms the conventional method MIRTK in terms of DICE score, but at the same time, it also demonstrates a higher folding ratio.

---

> > ### Comment · Reviewer_8syk · 2024-03-25
> > **the effect of large batch sizes**
> >
> > >we have not detected any detrimental effects [of the large batch size] on the optimization process
> >
> > While I do not think it is a big enough concern to downgrade my rating of this manuscript, the above point should be carefully investigated in follow-up work. Running the IDIR code on the DIRLAB set with 100k instead of 10k points for 2500 iterations, I do observe some performance degradation (mean error ~1.1mm @10k, ~1.3mm @100k batch size), despite the many more "examples" seen by the network. Hence, while it is possible that this had no impact in your dataset and experimental setting, I would not be surprised if this has a relevant effect after all.
> >
> > It _is_ possible I'm just seeing random variations. I only ran this as a quick experiment with a single random seed to check if my intuition made sense. I would say it warrants further investigation, though.

---

### Official Review · Reviewer_kLTk · 2024-02-29

**Confidence:** 4
**Preliminary Rating:** 4
**Recommendation:** Oral
**Final Rating:** 5

**Summary:**

This paper presents a new registration method using implicit neural representation for free-form deformations. The motivation of parameterizing the continuous deformable transformation is to reduce foldings. The effectiveness of this method is demonstrated through evaluations on the CamCAN dataset, where it shows promising outcomes.

**Strengths:**

- The motivation and the research question are clear.
- The paper addresses an interesting topic that is relevant to the MIDL community.
- The paper is easy to understand and follows a logical structure.
- The proposed extension is simple, straightforward, but clever and shows good results.
- The code is publicly available.

**Weaknesses:**

- The paper could be improved by giving a few more details about the method to make it easier for the reader to understand it. (see detailed comments).
- The authors do not compare the runtimes of the different algorithms. These are of course highly dependent on the available hardware, but give an indication of the speed.
- The authors do not evaluate their method on a challenge dataset, which would make it even easier to compare the method with state-of-the-art methods.
- The limitations of the proposed method could have been discussed in more detail.

**Detailed Comments:**

•	“we split into 80% training, 10% validation and 10% test set.” For the training of the DL models or also for the presented method?
•	“We choose the hyperparameters that achieve the highest Dice score for every method while not surpassing this folding ratio threshold.” This is performed on the training, validation or test dataset?
•	For the mono-modal registration 10k and the multi-modal registration 3.5M coordinates are used. It could be a nice addition to add to how many grid-points that would corresponde to.
•	“For the dataset, optimal outcomes are observed when the control points are spaced at 2mm3.” For both mono-modal and multi-modal registration? Maybe I am doing the maths wrong but it only corresponce to the 3.5M with a grid of 152x152x152.
•	“every 50 steps and performing an early stopping if the folding ratio becomes larger than 0.9%.” Is it then just stopped or rerun with different initialization or different parameter? Is the regularizer weight increased? If so how? The name stopping criteria gives the impression that you actually stop after that number of iteration and use that result.
•	How is different runtime using a batch size of 10k and 3.5M?

**Justification Of Final Rating:**

The rebuttal has addressed the open points and the manuscript has been improved. There are no further points for discussion. I have raised my ranking to Strong accept as I would like to see it presented at the MIDL conference. It is a very good and solid paper, but nothing ground-breaking.

**Justification Of The Preliminary Rating:**

The paper is of interest to the MIDL community. It proposes an interesting method, is well written and structured. The experiments are appropriate. It could be improved by adding more details to make it easier to understand and a more detailed discussion.

**Questions To Address In The Rebuttal:**

-	Please clarify open questions to make it easier to understand the method.
-	Please add the runtime of the method.
-	If possible (space-wise) add a discussion on the limitations of the proposed method. This could also be done in the experiments. Are there cases where the method doesn’t work well?

**Special Issue:**

Yes

---

> ### Author Response · Authors · 2024-03-17
>
> We would like to thank Reviewer kLTk for providing insightful feedback and highlighting inconsistencies in our manuscript. This constructive critique has prompted us to revise and enhance the clarity of these paragraphs. We address all concerns in the subsequent rebuttal and have revised the manuscript accordingly.
>
> >“we split into 80% training, 10% validation and 10% test set.” For the training of the DL models or also for the presented method?
>
> >[...] hyperparameters that achieve the highest Dice score [...] This is performed on the training, validation or test dataset?
>
> Deep learning CNN networks for registration, such as V-Morph and MIDIR, are trained on a cohort basis, whereas conventional methods like MIRTK and INR-based approaches are fitted on a pair of subjects. We opted to split the CamCAN dataset into 80% for training, 10% for validation, and 10% for testing. Subsequently, we trained the CNNs on the training set and fine-tuned their hyperparameters using the validation set. The test set evaluated the networks' ability for accurate and robust registration. Unlike CNN networks, pairwise methods do not require a training/validation/test split, and their hyperparameters can be tuned pairwise. Nevertheless, for comparative purposes against CNN methods, we calibrated the hyperparameters on the validation set and evaluated registration performance using the same test set as mentioned earlier.
>
> >[...] For both mono-modal and multi-modal registration? Maybe I am doing the maths wrong but it only corresponce to the 3.5M with a grid of 152x152x152.
>
> > For the mono-modal registration 10k and the multi-modal registration 3.5M coordinates are used. It could be a nice addition to add to how many grid-points that would corresponde to.
>
> We would like to thank the reviewer for raising the question regarding the grid size corresponding to 3.5 million points, as it prompted us to revisit our calculations and identify an error. As stated in the manuscript, our sampling entails two points in every direction at intervals of 2 mm³, resulting in a grid of (192/2)^3 = 884736 points (~890k, ⅛ of all points) for both mono and multi-modal registration, not the erroneously reported 3.5M points (½ of all points). We confirm that all experiments were conducted with the correct grid size of 884736 points, and the incorrect number was solely a mistake in reporting. Thank you for pointing this out!
> Moreover, since the grid spacing is 1mm^3, the number of sampled coordinates corresponds to the number of points we use.
>
> >[...] early stopping if the folding ratio becomes larger than 0.9%.” Is it then just stopped or rerun with different initialization or different parameter? [...] you actually stop after that number of iteration and use that result.
>
> Through empirical observation, we noted that the folding ratio initially begins at a low level during the fitting of the INR and significantly increases as image matching improves. Baseline INRs are more susceptible to this phenomenon than the proposed FFD-enhanced SINR. We fine-tune the hyperparameters to achieve a low folding ratio and a high dice score to ensure realistic deformation fields and attain high-quality registration results. During testing, we implement early stopping by setting a threshold of 0.9% for the folding ratio to guarantee the realism of the resulting transformations.
>
> >Please add the runtime of the method.
>
> We have included an indicative runtime table in the Supplementary Material [Table 2], and feel that this brings value to our manuscript.
>
> >How is different runtime using a batch size of 10k and 3.5M?
>
> In our experiments, the results obtained from small batch size training are generally more favorable, which is why we decided to compute the ReLU baselines for the optimal small batch sizes. The run-time may range from 45s (baseline without NMI) to1min39s (SINR with NMI).
>
> > If possible (space-wise) add a discussion on the limitations [...] Are there cases where the method doesn’t work well?
>
> Due to space constraints, we would like to clarify these ideas in the following section:
> Empirical Stopping Criteria: In the context of reconstruction tasks, INRs are typically trained to faithfully represent signals in the weights of a network, and the loss regarding the INR’s predictions/output only serves as a poor indicator to determine the faithfulness of the reconstruction in interpolation applications. In registration, this is even intensified, as we do not have a ground truth deformation field.
> While SINR allows us to successfully use INRs in the context of multi-modal, deformable image registration, it approaches the oscillation problem from a post-processing perspective and tries to address the folding issues within INRs.
> It would be interesting to see if regularization methods e.g. in the INR weight space, could allow to mitigate the oscillations, as SIREN has been suspected to be the cause of these problems.

---

> > ### Comment · Reviewer_kLTk · 2024-03-20
> >
> > I would like to thank the authors for the clarification. The rebuttal has addressed the open points and the manuscript has been improved. There are no further points for discussion.

---

### Meta-Review · Area_Chair_FnUC · 2024-04-02

**Recommendation:** Accept (Oral)
**Confidence:** 5

**Metareview:**

This is a very well-written paper which proposes a new method that is general and will likely have impact on registration tasks. The reviewers are unanimous in their appreciation of the paper and find the results interesting.

---

### Decision · Program_Chairs · 2024-04-06

Accept (Oral)